# Mean-Shift Distillation for Diffusion Mode Seeking

## Abstract

We present *mean-shift distillation*, a novel diffusion distillation technique that provides a provably good proxy for the gradient of the diffusion output distribution. This is derived directly from mean-shift mode seeking on the distribution, and we show that its extrema are aligned with the modes. We further derive an efficient product distribution sampling procedure to evaluate the gradient.

Our method is formulated as a drop-in replacement for score distillation sampling (SDS), requiring neither model retraining nor extensive modification of the sampling procedure. We show that it exhibits superior mode alignment as well as improved convergence in both synthetic and practical setups, yielding higher-fidelity results when applied to both text-to-image and text-to-3D applications with Stable Diffusion.

## 1. Introduction

Soon after image diffusion (Dhariwal & Nichol, 2021) models exploded in popularity, Poole et al. (2022) introduced the idea of using them for image optimization. Intuitively, this can be expressed as the notion that images more likely to be generated by a diffusion model are "better" in the sense of being more faithful to the data distribution the diffusion model was trained on.

Formally, diffusion models provide a mechanism to sample images $x \in \mathcal{I}$ from some learned distribution $p(x)$. We then have a parameter vector $\vartheta \in \mathcal{P}$, along with an image-generating model $g : \mathcal{P} \to \mathcal{I}$. Given an initialization $\vartheta^0$, we seek to optimize a $\vartheta^k$ such that $p(M(\vartheta^k)) > p(M(\vartheta^0))$. We expect this to yield an image $M(\vartheta^k)$ of higher quality, under the metric the diffusion model is trained for.

We could imagine optimizing $\vartheta$ by determining the gradient

---
[1]Anonymous Institution, Anonymous City, Anonymous Region, Anonymous Country. Correspondence to: Anonymous Author <anon.email@domain.com>.

Preliminary work. Under review by the International Conference on Machine Learning (ICML). Do not distribute.

$\nabla p(M(\vartheta))$ and ascending along it. However, while we may use our diffusion model to sample from $p(x)$, we can neither easily evaluate $p(x)$ nor determine its gradient $\nabla_x p(x)$. Even though we can formally express $p(x)$ in terms of the score function $\epsilon(x, t)$ through the instantaneous change of variable formula (Grathwohl et al., 2019), evaluating this formula requires calculating the divergence of the score function along the entire ODE path, making this of only theoretical interest. Evaluating the gradient of this quantity is even less practical.

Score distillation sampling (SDS) (Poole et al., 2022) attempts to address this problem by offering proxies for the density gradient that are easier to estimate. However, their theoretical properties are not rigorously established, and SDS suffers from significant bias as well as variance, yielding inaccurate gradients. Examining the loss landscape of SDS in Figure 1, we indeed see that not only are the maxima of this function not collocated with the modes of $p(x)$, but even in the simplest cases the loss creates "phantom modes" that are well out of distribution.

Our method offers both better alignment with the distribution and lower variance of the gradient estimate.

**Contributions.** In this paper, we propose *mean-shift distillation*, a distribution-gradient proxy based on a well-known mode-seeking technique. Furthermore, we show that:

- This proxy can be implemented easily, with minimal changes to the diffusion sampling procedure;

- It evaluates with less variance than SDS with improved mode alignment;

- It has superior behavior, converging to modes of the trained distribution with a clear termination criterion.

## 2. Related Work

**Denoising diffusion.** In our work we rely most directly on the mean-shift method of mode seeking (Cheng, 1995; Comaniciu & Meer, 2002), but our ability to apply it to diffusion rests on a body of theoretical analysis of this process.

Mathematically, denoising diffusion consists of solving an initial value problem (IVP) on a random variable from a

simple, typically standard normal distribution, where the time-dependent gradient is learned by reversing the process of adding noise to the distribution being modeled (Song et al., 2021b;a). Already in these works authors suggest ways in which the output distribution may be manipulated by adding terms to the differential equation underlying the initial value problem, a property we will rely on to manipulate the output to our method's advantage.

A surprising connection between mean shift and diffusion emerged from the analysis of the optimal denoising model (Karras et al., 2022; Chen et al., 2024). Since the forward (noising) process can be expressed as successive convolutions with a Gaussian kernel, the intermediate distributions are in fact Gaussian-kernel density estimates of the data distributions, with kernel bandwidth proportional to the time parameter. Therefore in the ODE of the reverse (inference) process, the gradient of the denoiser is theoretically equal to the mean-shift vector with appropriate kernel and bandwidth. Mean-shifting on the IVP time domain does not in fact seek modes of the output distribution, but we take advantage of this knowledge to implement mean shift on that domain.

Further related to the analysis of modes in particular, Karras et al. (2024); Bradley & Nakkiran (2024) suggest that applying classifier-free guidance (CFG) (Ho & Salimans, 2021) to diffusion has the effect of sharpening the modes of the output distribution. This guidance does not explicitly *seek* modes, but we have found that using CFG synergizes well with both SDS and our methodß.

**Distilling diffusion priors.** Score distillation sampling (SDS) (Poole et al., 2022; Wang et al., 2022) has emerged as a useful technique for leveraging the priors learned by large-scale image models beyond 2D raster images. SDS provides an optimization procedure to estimate the parameters of a differentiable image generator, such that the rendered image is pushed towards a higher-probability region of a pre-trained prompt-conditioned image diffusion model. Originally proposed to optimize volumetric representations like NeRFs, it has been extended to other non-pixel-based representations (Jain et al., 2023; Yi et al., 2024; Bahmani et al., 2024; Thamizharasan et al., 2024).

The tendency of SDS to produce over-smoothened results due to high variance is well documented. A plethora of works have been proposed to mitigate this behavior, e.g. to factorize the gradient to reduce the bias (Hertz et al., 2023; Yu et al., 2024; Katzir et al., 2024; Alldieck et al., 2024), or to replace the uniform noise sampling in SDS with noise obtained by running DDIM inversion Liang et al. (2023); Lukoianov et al. (2024). Wang et al. (2023a) propose a control variate for SDS, Wang et al. (2023b); Xu et al. (2024); Yan et al. (2025) improve diversity of generations, and Wang

et al. (2024) alleviate the multi-view inconsistency problem.

## 3. Mean-Shift Distillation

In this section we derive the mean-shift vector for the diffusion output distribution, and show how it approximates the gradient thereof. We further show how an efficient estimate of this vector may be obtained with a minimal modification of diffusion sampling. We begin with a motivation of our development by illustrating the pitfalls of SDS.

### 3.1. Motivation

Given a pre-trained diffusion model $\epsilon_\phi$, the SDS loss penalizes the KL-divergence of a unimodal Gaussian distribution centered around $x$ and the data distribution $p_\phi(z_t; y, t)$ captured by the frozen diffusion model conditioned on text embeddings $y$. With $x = g(\vartheta)$, an image rendered by $\vartheta$ via a differentiable renderer $g$, Poole et al. (2022) derive the gradient of the loss $\mathcal{L}_{\text{SDS}}$ with respect to $\vartheta$:

$$\nabla_\vartheta \mathcal{L}_{\text{SDS}} = \mathbb{E}_{t,\epsilon}\left[\alpha(t)\left(\epsilon_\phi(\alpha(t)x + \epsilon; t) - \epsilon\right)\right]\frac{\partial x}{\partial \vartheta},$$

$$\text{with } t \sim U(0, T), \epsilon \sim \mathcal{N}(0, \sigma(t)I). \quad (1)$$

To illustrate the pitfalls of SDS, we simulate it in 2D using a small denoising diffusion network (Figure 1). This allows us to set $\vartheta = x \in \mathbb{R}^2$ (where $g$ becomes an identity map). We construct a fractal-like dataset as shown by Karras et al. (2024), with analytic ground-truth probability density and score. This data distribution is a mixture of highly anisotropic Gaussians, where most of the probability mass resides in narrow regions, emulating the low intrinsic dimensionality of natural images (Roweis & Saul, 2000; Belkin & Niyogi, 2003). For a baseline, we compare it with DDIM (Song et al., 2021a), a popular first-order sampling algorithm, with classifier-free guidance (CFG) (Ho & Salimans, 2021). More details can be found in Section 4.2.

It is immediately apparent how even in this simple setting, the optima to which SDS converges do not model the output distribution well. Furthermore, the convergence itself is problematic due to very high variance of SDS, which we will address later.

### 3.2. Mean-shift Gradient Approximation

We start by convolving the data density $p$ with a radial Gaussian kernel $G_\lambda(x) = c_\lambda e^{-x^2/\lambda^2}$ with bandwidth $\lambda$, normalized by a constant $c_\lambda$. This convolution yields a smoothed density $p_\lambda^*(x)$:

$$p_\lambda^*(x) = p * G_\lambda(x) = \int G_\lambda(x - y)p(y)dy. \quad (2)$$

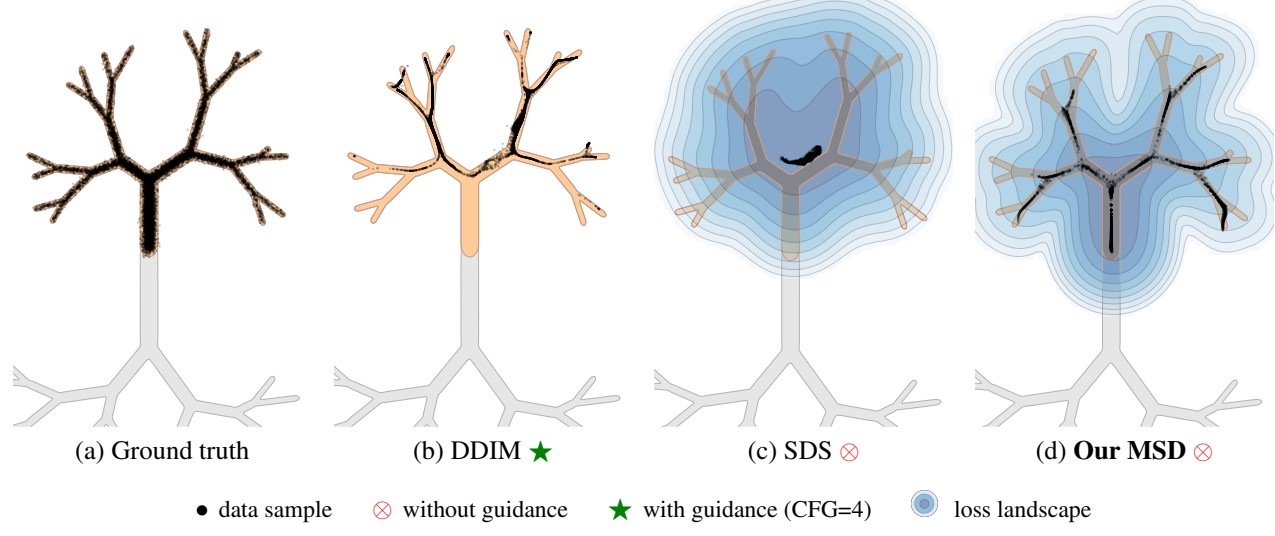

Figure 1: Mode-seeking simulated in a fractal-like 2D distribution with two (orange, gray) classes, adapted from Karras et al. (2024). We compare the behavior of diffusion sampling (DDIM) to optimization-based diffusion distillation, in a class-conditional setting. With class=orange, **(a)** Ground truth distribution, **(b)** DDIM sampling, **(c)** SDS optimization without guidance, and **(d)** our MSD optimization without guidance.

We now take the gradient $\nabla_x p_\lambda^*(x)$ of the smoothed density and substitute the Gaussian kernel's gradient:

$$\nabla_x p_\lambda^*(x) = \int \nabla_x G_\lambda(x-y)p(y)dy \quad (3)$$

$$= \int c_\lambda(x-y)G_\lambda(x-y)p(y)dy. \quad (4)$$

We then take the stationary-point equation and reorganize it as a fixed-point iteration:

$$\nabla_x p_\lambda^*(x) = 0 \implies x' = \frac{\int y G_\lambda(x-y)p(y)dy}{\int G_\lambda(x-y)p(y)dy}, \quad (5)$$

where the constant $c_\lambda$ cancels out. We will discuss the practical estimation of the integrals in Section 3.3 below.

The iterative process in Equation (5) is a continuous version of mean shift (Comaniciu & Meer, 2002). We may turn this into gradient proxy with several desirable properties. Defining the mean-shift vector $\vec{m}(x) = x' - x$, it follows from Equation (3) that

$$\vec{m}(x) \propto p_\lambda^*(x)\nabla_x p_\lambda^*(x). \quad (6)$$

Since the smoothed density $p_\lambda^*(x)$ is always non-negative, $\vec{m}(x)$ is always aligned with its gradient $\nabla p_\lambda^*(x)$. It is also aligned with the gradient of the true density $p$ as $\lambda \to 0$ (when such gradient exists). This means that a differential step along the vector $\vec{m}(x)$ will improve the likelihood of $x$, making this a good proxy for the kernel density estimation gradient. Furthermore, it implies that $\vec{m}(x)$ will be zero at the modes of $p_\lambda^*(x)$, giving us a convergence criterion.

### 3.3. Gradient Estimation via Product Sampling

The integrals in Equation (5) can be both estimated using samples $y$ from the density $p$; such estimation yields the classical mean-shift expression

$$x' = \frac{\sum_{y_i \sim p} K_\lambda(x-y_i)y_i}{\sum_{y_i \sim p} K_\lambda(x-y_i)}. \quad (7)$$

In our case, we do not have such samples readily available. We could in theory use images from the training dataset as these samples, or else use the diffusion model to generate them – either as a pre-process or on-the-fly during iteration. Unfortunately, that would be prohibitively costly as the datasets are typically quite large and accurate estimation would require a very large number of samples for practical (i.e. small) kernel bandwidths $\lambda$.

Our key insight is that the right-hand side of Equation (5) can be viewed as an expectation with respect to a density $\dot{p}_\lambda$ that is the product of $p$ and the kernel $G_\lambda$ centered at $x$:

$$x' = \int y \, \dot{p}_\lambda(y|x)dy = \mathbb{E}_{y \sim \dot{p}_\lambda(y|x)}[y]. \quad (8)$$

To generate samples $y$ from this product density, we exploit the fact that diffusion models employ score-based sampling (Song et al., 2021b; Dhariwal & Nichol, 2021). Instead of using the score $\nabla \log p$ of the density $p$ in DDIM sampling, we use the score of our product density:

$$\nabla_{z_t} \log(\dot{p}_\lambda(y|x)) = \nabla_{z_t} \log p(z_t) + \nabla_{z_t} \log G_\lambda(x-z_t)$$

$$= \nabla_{z_t} \log p(z_t) - \frac{x-z_t}{\lambda^2}, \quad (9)$$

which is the sum of the density score (provided by the diffusion model) and the score of our Gaussian kernel.

Having the ability to generate samples $y_i$ from the product density, we can estimate the mean-shift iterate (8) as

$$x' \approx \frac{1}{N} \sum_{y_i \sim \dot{p}_\lambda(y|x)} y_i. \tag{10}$$

In practice we use a single sample $y$, which simplifies our mean-shift vector to

$$\vec{m}(x) = y - x. \tag{11}$$

We can thus step along $\vec{m}$ to seek the modes of the data density $p$. Substituting a learned score model into Equation (9) gives us

$$\hat{\epsilon}_t = \epsilon_\theta(z_t; t) - \frac{x - z_t}{\lambda^2}. \tag{12}$$

### 3.4. Practical Considerations

**Impact of guidance.** Conditional score estimates from diffusion models, $\epsilon_\theta(z_t, c) \approx -\sigma_t \nabla_{z_t} \log p(z_t|c)$, are improved in practice with classifier-free guidance (CFG) (Ho & Salimans, 2021), which sharpens the distribution around the modes:

$$\tilde{\epsilon}_\theta(z_t, c) = (1 + w)\epsilon_\theta(z_t, c) - w\epsilon_\theta(z_t). \tag{13}$$

We may directly substitute this for the denoiser term in Equation (12). Despite its practical success, the denoising direction induced by CFG does not provide theoretical guarantees in producing samples from $p_{0,w}(z_t|c)$ (Bradley & Nakkiran, 2024). Even in simple settings, as observed in Figure 1(b), CFG can lead to mode drops. While alternative guidance strategy exists (Karras et al., 2024), we stick with the dominant practice of using CFG (Equation (13)). We have found that this synergizes well with mode-seeking by mean-shift, and show the effects of this in evaluation below.

**Integrating kernel score.** Because the magnitude of the kernel term in Equation (12) can be quite high when $|y - z_t|$ is high relative to $\lambda$, directly implementing this can result in instability while denoising particularly with explicit integrators. Higher-order integrators are generally capable of dealing with this instability, but require many more score function evaluations.

To address this, we note that in isolation the kernel term has the form of a negative exponential centered on $y$, or explicitly:

$$z_{t+\Delta t} = y + (z_t - y)e^{\frac{\Delta t}{\lambda^2}}, \tag{14}$$

where $\Delta t$ is negative. We take advantage of this to formulate a stable approximation that avoids the stability issues with a minimal change to the integration process. Instead of

feeding the full composite score function to the integrator, in each time step we first integrate only the score function with the existing integrator to get $z'_{t+\Delta t}$. Immediately after, we separately account for the kernel term by computing the final output as

$$z_{t+\Delta t} = y + (z'_{t+\Delta t} - y)e^{\frac{\Delta t}{\lambda^2}}. \tag{15}$$

We note such numerically instability has been observed when using high CFG values. A remedy is to apply guidance in a limited interval (Kynkäänniemi et al., 2024). We leverage similar ad-hoc tricks by applying the kernel term in limited interval through the sampling chain.

## 4. Practical Implementation and Evaluation

In this section, we construct synthetic examples on which we demonstrate that our proposed method behaves as theory predicts, alleviating the issues SDS exhibits even in these simple scenarios. We further explain the issues encountered when translating this theory into practice, and describe adaptations we designed to make our method work with real-world diffusion models. Finally, we evaluate that in these real-world scenarios, we retain the desirable properties and performance of our method. When applicable, we make comparisons with SDI (Lukoianov et al., 2024), a method that improves upon SDS by proposing a better noise term, yet, retaining the same gradient computation.

### 4.1. Idealized Setting

In order to manage large data dimensionality as well as massive training datasets, diffusion in practice employs a trained neural network to represent the denoiser $D$. However, Karras et al. (2022) have identified an analytical solution to minimizing the denoiser error, the *ideal denoiser* $D^*(x; t)$:

$$D^*(x; t) = \frac{\sum_i y_i \mathcal{N}(x; y_i, \sigma(t))}{\sum_i \mathcal{N}(x; y_i, \sigma(t))}, \tag{16}$$

where $y_0 \ldots y_n$ are samples in our training set. Attentive readers will notice that this is in fact the discrete mean shift formula (Comaniciu & Meer, 2002), with training samples taking the place of data samples and noise magnitude $\sigma(t)$ taking place of the kernel bandwidth $\lambda$. This expression is feasible to compute in practice for small datasets, and we may substitute it into the SDS formula (1) to get an explicit solution for the SDS gradient:

$$\nabla_x \mathcal{L}_{\text{SDS}} = \mathbb{E}_{t, z_t \sim \mathcal{N}(\alpha_t x, \sigma_t^2 \mathbf{I})} \left[ w(t) \frac{z_t - D^*(z_t; \sigma_t)}{\sigma_t} \frac{\partial x}{\partial \theta} \right].$$

This gives us an integral we can numerically evaluate with reasonable guarantees of its behavior. This allows us to compare both methods on synthetic datasets, eliminating any error introduced by training and evaluating a neural model to show that the theoretical properties hold.

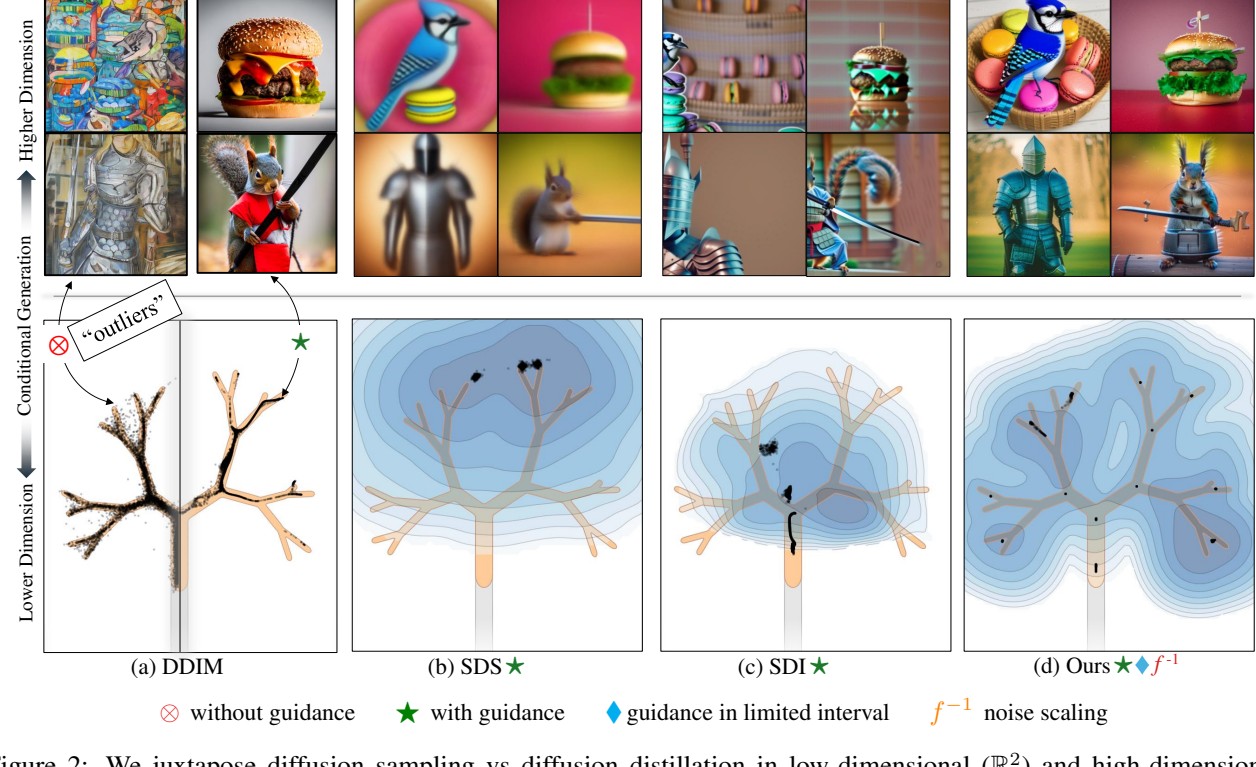

Figure 2: We juxtapose diffusion sampling vs diffusion distillation in low-dimensional ($\mathbb{R}^2$) and high-dimensional ($\mathbb{R}^{64\times64\times4}$) setting. **Top**: **(a)** text-conditioned generation of image via DDIM with 32 steps, **(b) - (d)** optimized coordinate-based neural implicit image for SDS, SDI, and our MSD respectively with StableDiffusion (CFG=7.5, § 4.4). **Bottom**: **(a)** class-conditioned generation of 2D points via DDIM with 32 steps, **(b) - (d)** optimized 2D points for SDS, SDI, and our MSD respectively (CFG=4, § 4.2). Text-prompts in clockwise order: *"A DSLR photo of a ... hamburger*, *squirrel dressed as a samurai weighing a katana*, *knight in silver armor*, and *bluejay on basket of macarons"*.

## 4.2. Toy Distributions in $\mathbb{R}^2$

In addition to the fractal dataset (Figure 1), we extend our analysis to other 2D datasets, with $y_i{}_{i=1}^{m} \subset \mathcal{M} \in \mathbb{R}^2$ sampled from various challenging toy 2D densities (van der Walt et al., 2014; Rozen et al., 2021). Given $10^4$ samples from the data distribution and points densely initialized across a grid $[-1.5, 1.5]^2$, we evaluate SDS (Algorithm 1) and our MSD (Algorithm 2) with $10^3$ Monte Carlo samples. We perform distillation with a learned denoiser (13) and an ideal denoiser (16). For the learned denoiser, we use the architecture and training setup used by Karras et al. (2024) and similarly represent the densities as mixtures of Gaussians. We use the Adam optimizer (Kingma & Ba, 2015) and run the optimization procedure for 150 steps with a learning rate of 0.08. For our MSD, we set an initial bandwidth of $0.316 \sim \sqrt{0.1}$ which is linearly decayed over the course of the optimization. For the ideal denoiser, due to it's high cost requirements in time and memory, we instead opt to use a few steps of gradient descent with high learning rate.

We visualize the generated samples produced by our method and SDS after the optimization in Figure 3. In addition

to optimizing samples, we evaluate both SDS and MSD gradients across the domain and then numerically integrate them to reconstruct the loss functions they represent. This makes the bias in SDS particularly obvious, as the peaks of this reconstructed function may be well out of distribution.

We suspect that this bias persists in SDS in higher dimensional settings and is what causes SDS optimized results to be blurry and exhibit other artifacts (top row of Figure 2).

In addition to bias, we are interested in evaluating the variance of the gradient estimate. This is an important factor for convergence, since ascending a stochastic estimate of the gradient is essentially a random walk. In such, high variance of the estimate may make the walk take longer to converge – indeed, with sufficiently high variance we may find the iteration often taking *backwards* steps with respect to the true gradient. Furthermore, a walk with high variance may not stay converged at an optima, and instead randomly oscillate around them.

To quantify the variance of an estimate $\hat{g}(x)$ of the gradient $g(x)$, we employ a slight variation of the Monte Carlo

Table 2: Metrics for 2D toy density datasets. *Left*: ideal denoiser ($D^*$) / *Right*: learned denoiser ($D_\theta$). MMD scaled by $10^{-4}$.

| Dataset | Method | NLL ↓ | Precision ↑ | Recall ↑ | MMD ↓ |
|---|---|---|---|---|---|
| Fractal | DDIM† | -1.85 / -1.51 | 0.97 / 0.95 | 0.93 / 0.96 | 0.86 / 0.007 |
| | SDS | 36.15 / 9.12 | 0.08 / 0.01 | 0.03 / 0.0 | 328.03 / 87.04 |
| | Ours | -1.32 / -2.02 | 0.92 / 0.97 | 0.33 / 0.42 | 30.46 / 12.79 |
| Spiral | DDIM† | -1.39 / -1.32 | 0.97 / 0.96 | 0.93 / 0.96 | 0.41 / 1.16 |
| | SDS | 30.37 / 8.13 | 0.02 / 0.04 | 0.03 / 0.11 | 13.85 / 274.35 |
| | Ours | -1.28 / -1.51 | 0.99 / 0.98 | 0.18 / 0.18 | 4.49 / 18.41 |
| Pinwheel | DDIM† | -1.19 / -1.1 | 0.97 / 0.97 | 0.94 / 0.97 | 1.05 / 0.27 |
| | SDS | 2.29 / 2.00 | 0.85 / 0.90 | 0.03 / 0.005 | 5.18 / 36.37 |
| | Ours | -1.94 / -2.19 | 0.99 / 0.99 | 0.01 / 0.13 | 5.83 / 7.25 |

estimator efficiency formula

$$\varepsilon(\hat{g}(x)) = \frac{|g(x)|^2}{\text{MSE}(\hat{g}(x))\,\text{cost}(\hat{g}(x))}. \quad (17)$$

We measure cost as number of invocations of the score model, since that is the typical bottleneck in diffusion. Normalization by the squared norm of $g$ is included to account for the fact that due to bias and scaling, different estimators may converge to gradients of different magnitude, and the normalized MSE then roughly describes the probability of the estimated gradient pointing the "right" way.

We accumulate both MSE and cost over many independent estimations, and average over many values of $x$. The result of these efficiency comparisons between our method and SDS is in Table 1 (in log-scale). Although getting a single estimate with our method requires more score model invocations, the efficiency of our method is significantly higher. We observe similar behavior in complex settings (Figure 4). In Table 2, we evaluate the quality of the generated samples. Our method outperforms SDS, in both the ideal denoiser and learned denoiser setting.

Table 1: Efficiency ↑

| Dataset | SDS | Ours |
|---|---|---|
| Fractal | -6.89 | 7.65 |
| Spiral | -7.57 | 6.32 |
| Pinwheel | -6.99 | 7.08 |

### 4.3. Practical Setting

For large-scale image datasets, idealized denoiser is no longer tractable and we have to contend with a learned denoising function, as well as all the associated machinery.

This introduces numerical issues. Namely, as already mentioned in Section 3.4, the magnitude of the kernel term may grow to where the standard first or second order integrators can no longer manage it; but conversely, so does the magnitude of the learned score term when $z_t$ is far out of distribution; why this is the case becomes apparent when we consider that the ideal denoiser (Section 4.1) uses the same equation as mean shift. This is a problem because at the start

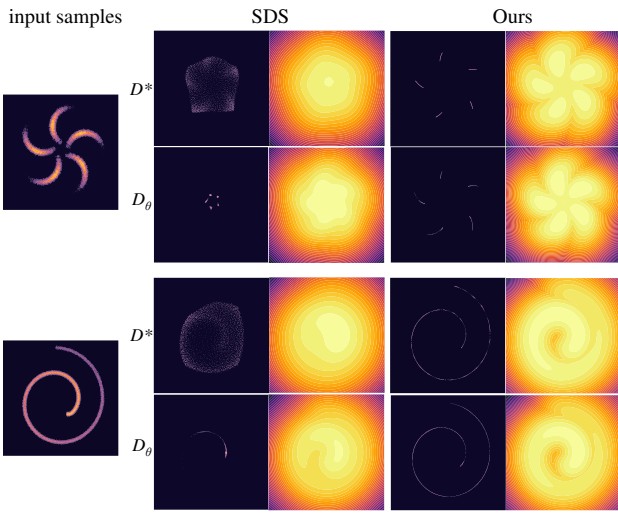

Figure 3: Unconditional distillation on two toy density datasets, given an ideal denoiser ($D^*$) and a learned denoiser ($D_\theta$). For each method and denoiser, we show the optimized samples (left) and the loss landscape (right). *Zoom in for clarity.*

of the optimization, it is likely in a high-dimensional space that $x$ will be out of distribution and we have to choose between the integration failing because the denoiser term has a high magnitude, or because the kernel term has a high magnitude.

To alleviate this, we use two heuristic approximations: applying guidance in limited interval (Section 3.4) and scaling our sample in Equation (15) by noise corresponding to time step $t$. In practice, we apply inversion to get the latter. These are designed to keep the iterate in a region with reasonable score magnitude and still sample a distribution that is an approximation of the product distribution.

### 4.4. Pre-trained Stable Diffusion

We use the latent-space diffusion model, Stable Diffusion, as the diffusion prior for text-conditioned optimization of parameters of differentiable image generators. Specifically, we optimize parameters $\vartheta$ of generator $g$, a rendering function that maps $\vartheta$ to an image $\mathcal{I}$. The rendered image $\mathcal{I}$ is fed to the image encoder to get $x^k$, our latent at optimization step $k$, over which the gradient is computed. We define two settings, (1) where $\vartheta$ represents an RGB image, and (2) where $\vartheta$ represents a 3D volume. Specifically:

1. **Text-to-2D.** We represent 2D images via a coordinated-based MLP $f$ with learnable parameters $\vartheta$ that takes as input a 2D point $p$ in the unit square $p = (x, y) \in [0, 1]^2$ and outputs RGB $\in [0, 1]^3$; $f(p; \vartheta) : \mathbb{R}^2 \to$ RGB. We use this non pixel-based representation of an image for two reasons, (1) to prevent our method and the baselines from taking the exact gradient step i.e.

Table 3: Text-to-2D quantitative comparison. We evaluate fidelity with FID and CLIP-SIM. [†]FID measured with DDIM as ground truth.

| Method | FID ↓ | CLIP-SIM (L/14) ↑ |
|---|---|---|
| DDIM[†] | - | $44.1 \pm 2.8$ |
| SDS | 198.90 | $27.7 \pm 1.9$ |
| SDI | 166.16 | $31.0 \pm 0.7$ |
| Ours | 114.12 | $32.6 \pm 0.8$ |

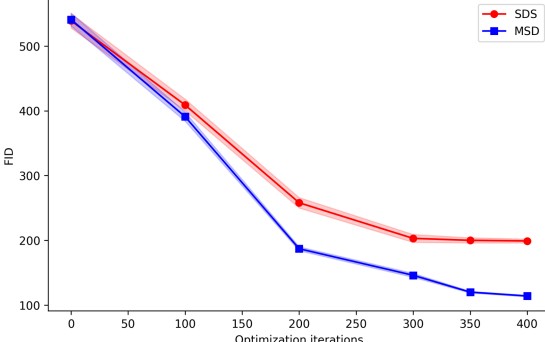

Figure 4: FID vs optimization iterations.

running diffusion sampling and setting $x^k$ to the denoised latent $z_0$, and (2) we can directly compare with images sampled via DDIM, an unconstrained image generation setting.

2. **Text-to-3D.** We represent 3D volumes as NeRFs, following (Poole et al., 2022). The NeRF is parameterized by two MLPs, one for foreground and one for background. The former has 64 hidden nodes and 2 layers, with input $(x, y, z)$ coordinates encoded via HashGrid (Müller et al., 2022).

**Implementation details.** We implement all our code in PyTorch, on a single NVIDIA A100 gpu. We use the Threestudio (Guo et al., 2023) framework for experiments involving pre-trained Stable Diffusion. All the experiments use Adam optimizer with lr= $10^{-2}$. We set optimization steps to 400 for text-to-2D and $7k$ for text-to-3D. We use a monotonically decreasing schedule for the bandwidth $\lambda$.

### 4.5. Evaluation

**Dataset.** We use a subset of the prompts curated by Poole et al. (2022); Hertz et al. (2023). We include all prompts in Appendix B.

**Metrics.** For toy density dataset (Section 4.2), we compute negative log-likelihood scores (NLL), generative precision and recall (Kynkäänniemi et al., 2019), and maximum mean discrepancy (MMD). For text-to-2D, we use images

produced by DDIM to represent the ground truth distribution. To evaluate fidelity of the images, FID (Heusel et al., 2017) is computed for each baseline (SDS, SDI) and ours against this ground truth image set. We also compute CLIP scores (cli) to measure prompt-generation alignment.

**Quantitative comparisons.** Table 3 reports results for FID and CLIP-based similarity, comparing our method with SDS and SDI. We outperform both baselines in image fidelity and are our generations are more faithful to the input prompt.

**Qualitative comparisons.** Figure 2 (top row) and Figure 8 compares our method with SDS and SDI on text-to-2D generation, qualitatively. In the latter, we show the importance of the two heuristics (Section 4.3) to resolve numerical instabilities, absence of which can result in visual artifacts. SDS, as discussed, produces low-fidelity results while SDI's inversion accumulates numerical errors during early stages of optimization. In Figure 5, we qualitatively compare results for text-to-3D optimization. We restrict to qualitative comparison for this task as quantitative metrics have high variance due to the absence of a ground truth dataset.

**Impact of bandwidth.** Figure 6 shows the impact of the bandwidth ($\lambda$) term on the denoising process. First, we sample three parameters $\{\vartheta^k\}$ from our text-to-2D optimization pipeline at iterations $k = \{100, 200, 400\}$ and also sample three discrete $\lambda$ values $\{\lambda_1 \ll \lambda_2 \ll \lambda_3\}$. Then, we run our forward pass once for each $\lambda_i$, independently. We visualize four decoded denoised latents $z_0$ (with different random seeds). The highlighted images show the optimal choices of $\lambda$ for each $x^k$ (the encoded latent for $\vartheta^k$). At high bandwidth value $\lambda_3$, the influence of the kernel term in the product sampling is negligible. This degenerates to vanilla denoising and we observe high variance in the output, irrespective of our current $x^k$. This is ideal at early stages of optimization. As bandwidth is annealed, we observe reduction in variance. Yet, the quality of the outputs can degrade if the kernel term dominates while $x^k$ is not *"in-distribution"*. As $x^k$ approaches the mode of the distribution corresponding to the input text-prompt at final stages of optimization (when $k = 400$), with a low bandwidth $\lambda_1$, our denoised latent $z_0 \approx x^k$. This provides us with a convergence criteria and we terminate when $\lambda$ is below the threshold $\lambda_1$.

## 5. Conclusion

In this paper, we have reframed diffusion distillation in terms of explicitly ascending the gradient of the data distribution. We have derived mean-shift distillation as a proxy that provably aligns with this gradient, and in the limit its maxima are collocated with the modes of the data distribution.

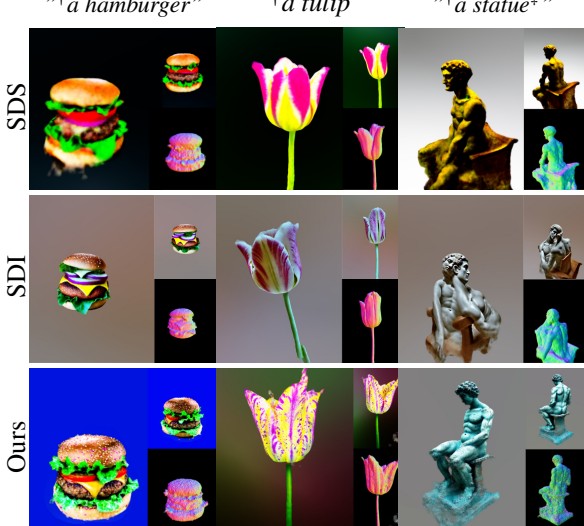

Figure 5: Comparison of 3D generation with other score distillation methods. Full prompt: [†]*"A DSLR photo of a ..."*, [‡]*"a Michelangelo statue of a man on a chair"*.

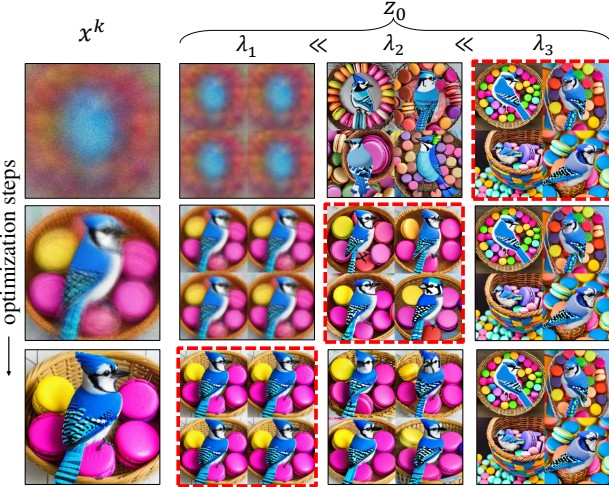

Figure 6: Impact of bandwidth ($\lambda$) on the denoised latent ($z_0$). We set $\lambda_3 = 10^3$, $\lambda_2 = 10$, $\lambda_1 = 10^{-2}$. Highlighted images show the optimal bandwidth value corresponding to the $k^{th}$ optimization.

We have demonstrated that compared to SDS, this method achieves better mode alignment as well as lower gradient variance, which in practice translates to more realistic optimization results as well as improved convergence rate.

Since this method simply provides optimization gradient much like SDS does, it may be used as a one-to-one replacement without retraining of the underlying model, or indeed substantial code modification.

While the basic algorithm works as the theory predicts in synthetic scenarios, with real-world models we have to contend with integrator error due to large score magnitudes. We have designed heuristics to alleviate this and achieve improvements on SDS in practice, but we hope future work will be able to improve the integration and/or sampling procedure, obviating the need for heuristics.

## Impact Statement

As a more or less straightforward substitute of an existing method (SDS), our method inherits ethical concerns of the diffusion models it is being applied to, and the applications it is being put towards. It remains important to take care with sourcing training data to avoid copyright issues, bias issues, and training harmful content into the model. On the output side, generative models improve accessibility to creative expression, which however also makes it easier to produce harmful content including, but not limited to, misinformation, defamatory and obscene images. Ultimately these issues are impossible to fully solve on the tooling side and we must rely on other methods to analyse content and establish authenticity thereof to compensate.

That said, improved convergence properties of our method mean that less computation is required to achieve the same result, alleviating some of the environmental impacts associated with these generative methods.

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

## A. Implementation details

**Input** : pre-trained diffusion model $\epsilon_\theta : \mathbb{R}^{d_1 \times \cdots \times d_k} \to \mathbb{R}^{d_1 \times \cdots \times d_k}$, target parameters $\psi \in \mathbb{R}^d$, condition $c$, mapping function $g(\psi) : \mathbb{R}^d \to \mathbb{R}^{d_1 \times \cdots \times d_k}$, time-dependent functions $w(t), \alpha(t)$, Monte Carlo sample size $N$.

**Output** : $\psi^*$

---

**Algorithm 1** Distillation via SDS

**for** $k = 1, \ldots, steps$ **do**
   $x^k \leftarrow g(\psi)$
   **for** $i = 1, \ldots, N$ **do**
      $t \leftarrow \mathrm{U}(0, 1)$
      $z_t \leftarrow \alpha(t)x^k + \epsilon_t$
      $y_i \leftarrow w(t)[\epsilon_\theta(z_t, t, c) - \epsilon_t]$
   **end**
   $\nabla_\psi \mathcal{L}_{SDS} \leftarrow \frac{1}{N} \sum (y_i - x^k)$

   `// Backpropagate` $\nabla_\psi \mathcal{L}_{SDS}$`, update` $\psi$
**end**

---

**Algorithm 2** Distillation via MSD (Ours)

**function** *ODESolver($x, \lambda$) (eq 12)*
   $z_T \leftarrow \mathcal{N}(0, I)$
   **for** $t = T - 1, \ldots, 0$ **do**
      $z_t \leftarrow \epsilon_\theta(z_t, t, c) - (x - z_t)/\lambda^2$

   **end**
   return $z_0$
**function** *ODESolver($x, \lambda$, stable) (eq 15)*
   $\{z_t^*\}_{t=0}^T \leftarrow inversion(x)$
   $z_T \leftarrow z_T^* + (\epsilon - z_T^*)e^{-\Delta t/\lambda^2}$
   **for** $t = T - 1, \ldots, 0$ **do**
      $z_T \leftarrow z_t^* + (\epsilon_\theta(z_t, t, c) - z_t^*)e^{-\Delta t/\lambda^2}$

   **end**
   return $z_0$

`// initialize` $\lambda$`, set` $\lambda_{min}$
**for** $k = 1, \ldots, steps$ **do**
   $x^k \leftarrow g(\psi)$
   **for** $i = 1, \ldots, N$ **do**
      $y_i \leftarrow \text{ODESolver}(x^k, \lambda)$
   **end**
   $\nabla_\psi \mathcal{L}_{MSD} \leftarrow \frac{1}{N} \sum_i^N (y_i - x^k)$

   `// Backpropagate` $\nabla_\psi \mathcal{L}_{MSD}$`, update` $\psi$
   `// Anneal` $\lambda$
   **if** $\lambda < \lambda_{min}$ **then**
      `// terminate`
   **end**
**end**

---

Figure 7: Pseudocode of SDS and our procedure, MSD. We additionally show the numerically stable solver, which is used for experiments with Stable Diffusion.

## B. List of prompts

   *"A DSLR photo of a hamburger"*

   *"A blue jay standing on a large basket of rainbow macarons"*

   *"A DSLR photo of a squirrel dressed as a samurai weighing a katana"*

   *"A DSLR photo of a knight in silver armor"*

   *"Line drawing of a Lizard dressed up like a victorian woman, lineal color"*

   *"A photo of a car made out of sushi"*

*"A DSLR photo of a tulip"*

*"A DSLR photo of a Pumpkin head zombie, skinny, highly detailed, photorealistic"*

*"A watercolor painting of a sparrow, trending on artstation"*

*"Michelangelo style statue of man sitting on a chair"*

## C. Ablations and more results

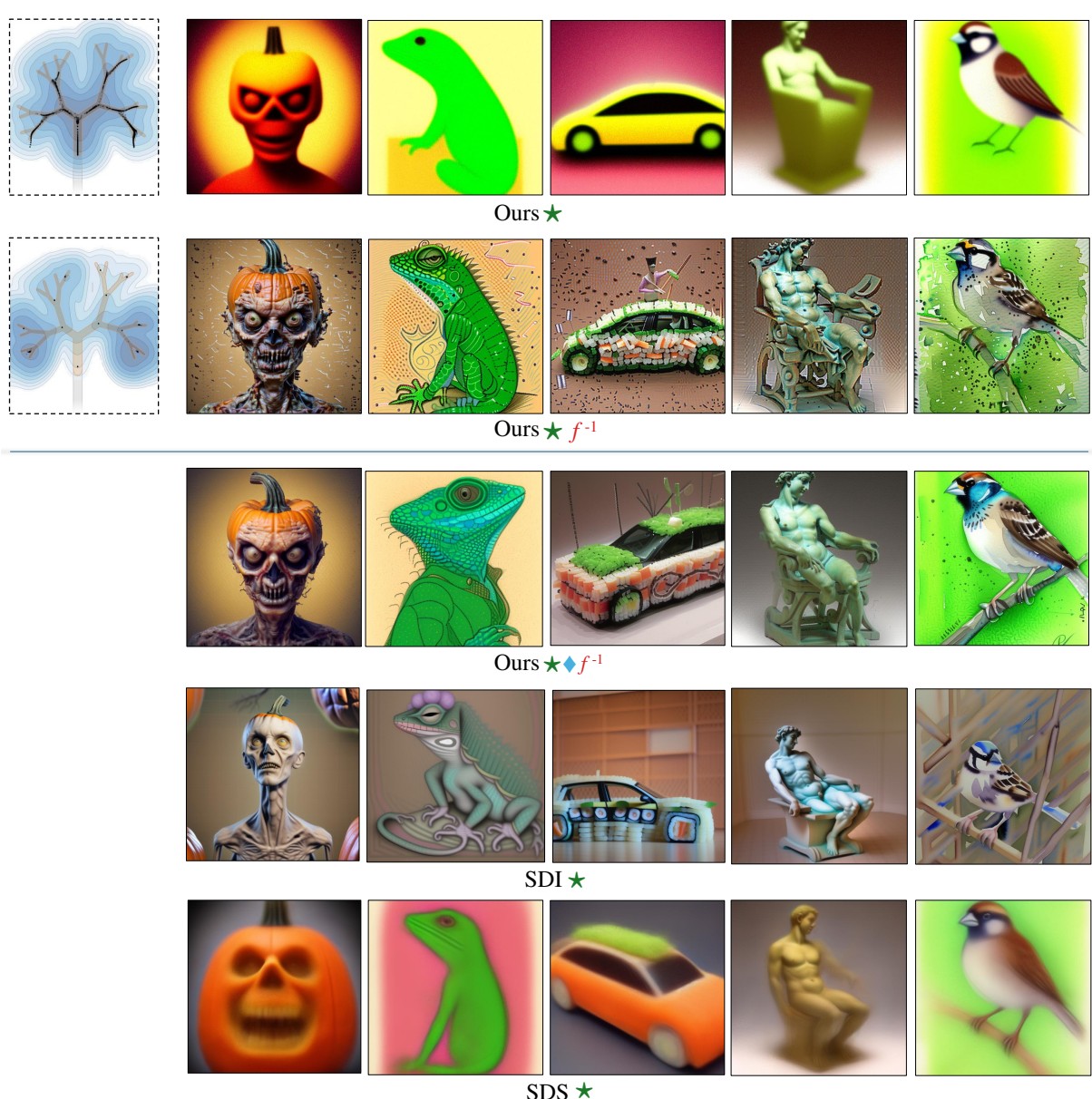

Figure 8: We extend Figure 2 with two ablations; applying guidance in the entire denoising trajectory (row 1) and noise scaled sample in the kernel term (row 2) (§ 4.3). We also show additional results for our method, SDI and SDS (row 3,4,5 respectively).

