# OpenReview forum: "Mean-Shift Distillation for Diffusion Mode Seeking"
_ICML.cc/2025/Conference — Submitted to ICML 2025_

### Official Review · Reviewer_6KVp · 2025-02-27

**Overall Recommendation:** 4

**Summary:**

The paper systematically explores behavior of the popular SDS algorithm for sampling of diffusion models and concludes that it does not well cover the true modes of the underlying distribution. Similar observations are made for another recent alternative - SDI. Inspired by the Gaussian paths defining training of diffusion models, the authors adapt the mean shift algorithm to express the gradient of probability function useful for optimization of the images. They proceed to demonstrate efficiency of the algorithm in recovering the modes of the distribution on both small synthetic datasets with known ground-truth and a real image denoiser from StableDiffusion. The method produces more stable and cleaner images than the altneratives which is also useful for tasks such as 3D reconstruction. Finally, the authors explore the parameter space to motive their design choices.

## update after rebuttal

The rebuttal has addressed my concerns. I recommend accentance because the paper states a clear, relevant and important problem (finding the modes), proposes well motivated and theoretically founded algorithm (MDS) and clearly shows that it achieves an improvement compared to a relatively recent work (SDI). Is also provides illustrative low-dimensional examples clearly visaulizing the algorithm behavior beyond just showing nice output images.

**Claims And Evidence:**

Yes. The authors claim this is a swap-in replacement for SDS which they demonstrate with their algorithm. They also achieve better distribution coverage in their experiments.

**Essential References Not Discussed:**

None.

**Experimental Designs Or Analyses:**

The design is in line with common practice and it is well explained.

**Methods And Evaluation Criteria:**

Yes, the method is well motivate. The design and evaluation of the experiments make sense and it is a good fit for the problem.

**Other Comments Or Suggestions:**

The formulation "gradient of the diffusion output distribution" from the abstract is a bit cumbersome.

Typos:
	- L84: methodß

**Other Strengths And Weaknesses:**

**Strengths**

+ Well written, well explained.
+ Very nice illustration of the problem and achieved results. I like the utilization of the exact denoiser for analysis (even though it is inspired by prior work). The method seems efficient in what it tries to achieve.
+ Good results.

**Weaknesses**

- While the results of SDS are well represented in the paper, SDI is not as widely shown. E.g. it is not displayed in tables 1 and 2 and Fig 3. Edit: Additional values provided in the rebuttal.
- The z_t in eq. 9 is understood to be the noisy version of x_t (or just noise in the limit) but I do not think it is properly defined in the paper.
Edit: A correction promised in the rebuttal.

**After rebuttal**

The rebuttal has addressed my concerns and I recommend acceptance.

**Questions For Authors:**

1) The 3D shapes in Figure 5 remain very highly saturated despite the method generally converging to the modes. Does it represent a quality aspect that cannot be well illustrated by the loss visualization in a low dimensional version of the problem?

2) Is the mode seeking behavior always desirable? Does it not come at a cost of diversity by avoiding less common but potentially interesting samples?

3) Figure 8 shows that the guidance in limited interval is important for the results. Would a similar trick also be effective for SDI?

4) How many samples per prompt were used in the experiments? FID computation typically requires thousands a few samples to avoid bias. [Bińkowski, Mikołaj, et al. "Demystifying MMD GANs." International Conference on Learning Representations. 2018.]

**Relation To Broader Scientific Literature:**

Yes, the authors discuss relevant literature clearly and efficiently. Establish connection to their work.

**Theoretical Claims:**

The mathematical derivation of the MDS algorithm seems correct as far as I can judge. I cannot see any errors.

---

> ### Author Rebuttal · Authors · 2025-03-31
>
> We thank the reviewer for the positive evaluation and for recommending acceptance. We are glad they recognize our contributions. Below, we answer questions raised in the review:
> > While the results of SDS are well represented in the paper, SDI is not as widely shown. E.g. it is not displayed in tables 1 and 2 and Fig 3.
>
> In our rebuttal to Reviewer cVvL we extend Table 1, 2, and 3 with comparisons to SDI and a new baseline VSD [1], where missing. We qualitatively compare with SDI on the fractal dataset in Fig. 2. We will also extend it, along with VSD, to the spiral and pinwheel dataset shown in Fig 3 in our revision.
>
> > The z_t in eq. 9 is understood to be the noisy version of x_t (or just noise in the limit) but I do not think it is properly defined in the paper.
>
> Thank you for highlighting this. We will define the term in our revision.
>
> > Is the mode seeking behavior always desirable?
>
> The desirability of mode seeking varies between applications. When trying to directly sample images from the trained model, it is true that we wish to sample from the full variety of the distribution instead of getting only the mode. Methods like DDIM aim for this.
> On the other hand, when we are optimizing an image (or using the image as a proxy to optimize, e.g. NeRF parameters), any gradient-based optimization will converge to a set of sparse points - local extrema - where the gradients are zero (if it converges at all). This is the intended use-case for SDS, SDI, and our method, and in this case, it is not possible in general to have the optimization process converge to a distribution of points. Given that, the best we can guarantee is that the points the process converges to are aligned with the distribution. Mode-seeking is our proposed way of achieving that.
>
> > Highly saturated results in Figure 5
>
> We have observed this “washed out” look in the image and 3D results, despite using a low guidance scale (CFG=7.5). We believe this to be an artifact of using CFG in the pipeline, because these effects appear in both our method and the baseline and are consistent with what has been observed in previous literature [2][3]. The analysis on the fractal toy dataset by Karras et al., which we also use in our paper, suggests that this is the visual effect of “sharpening” the modes of the distribution that can also be observed in these 2D test cases.
>
> > Figure 8 shows that the guidance in limited interval is important for the results. Would a similar trick also be effective for SDI?
>
> SDI only performs one denoising step in each optimization iteration. It is not straightforward to apply guidance in a limited interval without substantial modifications to their algorithm.
>
> > How many samples per prompt were used in the experiments
>
> For each method, we generate 500 samples per prompt. With a total of 10 prompts, this gives us a total of 5,000 images per method.
>
> References:
>
> [1] Wang Z, Lu C, Wang Y, et al. “Prolificdreamer: High-fidelity and diverse text-to-3d generation with variational score distillation”. Advances in Neural Information Processing Systems, 2023.
>
> [2] Ho and Salimans, “Classifier-Free Diffusion Guidance”, 2021.
>
> [3] Saharia et al., “Photorealistic Text-to-Image Diffusion Models with Deep Language Understanding”, 2022.

---

### Official Review · Reviewer_ZAGu · 2025-03-17

**Overall Recommendation:** 1

**Summary:**

This paper presents mean-shift distillation, a diffusion distillation technique that provides a provably good proxy for the gradient of the diffusion output distribution.

**Claims And Evidence:**

The claims made in the submission are supported by clear evidence. However, the evidence is not convincing enough. For example, only one dataset is evaluated. It will be better if more datasets can be evaluated. In addition, the used prompts are limited.

**Essential References Not Discussed:**

The compared methods, including FID and CLIP-based similarity, are not discussed in the related work.

**Experimental Designs Or Analyses:**

Compared methods are not sufficient. Besides FID which was published in 2017, the latest baselines should be compared. The paper on CLIP-based similarity was not cited.

**Methods And Evaluation Criteria:**

The proposed method and evaluation criteria make sense for the problem.

**Other Comments Or Suggestions:**

In "Distilling diffusion priors" of Section 2, it would be better if the limitations of the related works could be discussed.

**Other Strengths And Weaknesses:**

1. Writing needs to be improved.
2. Evaluation is not comprehensive.

**Questions For Authors:**

N/A

**Relation To Broader Scientific Literature:**

The key contributions of the paper are relatively related to the broader scientific literature as it is better than FID and CLIP-based similarity.

**Theoretical Claims:**

1. In eq. (1), the meaning of $t$ and $\alpha(t)$ are missing.
2. In eq. (2) - eq. (5), it is clear that $p$ is the data density, but the meaning of $p(y)$ is missing.
3. In eq. (7), the meaning of $K_{\lambda}$ is missing.

---

> ### Author Rebuttal · Authors · 2025-03-31
>
> We thank the reviewer for the feedback. We hope the following will resolve any confusion regarding our contribution and evaluations:
>
> - We would like to emphasize that the goal of our work is to improve mode seeking and achieve better distillation for diffusion models, not to improve metrics for image generation, such as FID, or metrics for evaluating image-text alignment, such as CLIP similarity. Neither FID nor CLIP-based similarity are baselines for our method, but rather the metrics by which we compare our method to the baselines (SDS, SDI, VSD), as in e.g. Table 3. FID and CLIP-based similarity are widely used evaluation metrics for image generative models in all state-of-the-art research works. We also cite the library we used (torchmetrics from PyTorch Lightning).
>
> - We would also like to point out that we evaluate on four datasets: three synthetic datasets demonstrating that current distillation methods fail even in toy distributions and the pre-trained text-to-image StableDiffusion-XL model for the practical setting. The latter was used in all our baselines (SDS, VSD, and SDI). Our text prompts are borrowed from DreamFusion (Poole et al., 2022).
>
> - Regarding the three points raised in the theoretical claim part of the review, we will clarify in the paper:
>
> (1) $t$ is the time step used in denoising diffusion, and $\alpha(t)$ is the time-dependent weighting function. Both these terms are standard notation in the denoising diffusion literature [1]. Eq. 1 is the SDS gradient from DreamFusion, where $t$ is randomly sampled.
>
> (2) $p(y)$ is the data density, and $p(x)$ is the smoothed data density after convolving the data density with the Gaussian Kernel.
>
> (3) $K_\lambda$ is the kernel used in mean-shift clustering, which determines the weight of nearby points for re-estimation of the mean. There have been various choices for kernels in mean-shift clustering, e.g., Gaussian, Epanechnikov, flat kernel [2]. Our analysis focuses on the case $K_\lambda (x)$ is the Gaussian kernel used in Eq. 2. In Eq. 7, we write the general expression for mean-shift updates for any kernel function.
>
> References:
>
> [1] Denoising Diffusion Probabilistic Models. J. Ho, A, Jain, P. Abbeel. NeurIPS 2020.
>
> [2] https://en.wikipedia.org/wiki/Mean_shift

---

### Official Review · Reviewer_1C1e · 2025-03-18

**Overall Recommendation:** 3

**Summary:**

This paper proposes a new diffusion distillation technique called mean-shift distillation (MSD), intended to solve the problem of mode-seeking when leveraging pre-trained diffusion models for tasks like text-to-2D or text-to-3D optimization. Existing approaches such as score distillation sampling (SDS) are known to suffer from high gradient variance and bias, often converging to sub-optimal or “out-of-distribution” modes. This paper aims to correct that by explicitly seeking the modes of the learned data distribution in a principled way.

The authors derive a gradient estimator based on classical mean-shift mode-seeking. They prove that the estimator is aligned with the gradient of the distribution’s density in an approximate or smoothed sense, which in theory leads to better mode alignment. Besides, a key technical idea is to sample from a product of the data distribution (modeled via diffusion) and a Gaussian kernel centered on the current iterate, thereby estimating the mean-shift vector with minimal extra overhead. The proposed approach can be dropped into existing SDS pipelines without retraining diffusion models. The authors implement a few heuristic stability strategies (e.g. limiting guidance to a restricted interval) to address numeric issues.

**Claims And Evidence:**

The main theoretical claim—that mean-shift distillation aligns with the gradient of the smoothed density—is a well-known property of mean-shift, and the authors adapt it cleanly for the diffusion setting. The improved mode alignment claim is supported by 2D toy experiments where the ground truth distribution is known and the authors can directly observe that SDS creates biased maxima whereas MSD recovers actual modes.

As for the real world text-to-2D and text-to-3D tasks, though more qualitative, support the claim that MSD produces sharper, more mode-aligned results (more faithful images, better 3D shapes, etc.) than SDS.

However, while the authors introduce heuristics to handle numeric instabilities in high dimensions, it remains somewhat ad hoc. The general statement that their approach is “stable with minimal changes” may need more thorough ablation on large-scale tasks to confirm its robustness.

**Essential References Not Discussed:**

There is no major omissions.

**Experimental Designs Or Analyses:**

The 2D toy experiments are carefully designed to highlight fundamental differences between SDS and MSD, especially focusing on distribution “phantom modes.” The approach is thorough and compelling for diagnosing method-level phenomena.
The text-to-2D and text-to-3D experiments with Stable Diffusion reflect the standard practice in the field.

The paper sketches the pseudo-code for MSD, clarifies some crucial heuristics, and references open-source frameworks (ThreeStudio). This should be sufficient for other researchers to replicate the approach. Some details about integrator stability, partial inversion, or bandwidth scheduling might need more elaboration for a fully plug-and-play experience, but overall the methodological details are clearly stated.

**Methods And Evaluation Criteria:**

The mean-shift-based approach is a well-established idea for mode-seeking in density estimation. Extending it into diffusion-based generative modeling by sampling from a product density is a neat conceptual step.

The evaluation focuses on synthetic 2D data (where the ground truth distribution is known) to check theoretical correctness, and then transitions to text-to-2D and text-to-3D tasks with a large pre-trained diffusion model (Stable Diffusion). This is a reasonable design to capture both theoretical clarity and real-world practicality.

However, in my opinion, although the chosen baselines in this paper (SDS, SDI, plus direct sampling like DDIM) are the most widely recognized, there are other methods including classifier guidance and variance reduction according to the authors' reference list. It would be more convincing if the authors can compare their methods with these baselines.

**Other Comments Or Suggestions:**

Please refer to the "Weaknesses" section.

**Other Strengths And Weaknesses:**

Strengths:
In this paper, the authors provided a clean, conceptually simple remedy (mean-shift) for improving the alignment of the distillation gradient with the distribution’s modes. The 2D analysis is carefully executed, showing the shortfalls of SDS and the better behavior of MSD. As for the practical implementation, there are minimal code changes and no model retraining required, which are quite impressive.

Weaknesses:
1. You note that the theoretical foundation for classifier-free guidance (CFG) is less rigorous, and the paper does not provide fully rigorous proofs on the possible side effects of classifier-free guidance combined with mean-shift—though it is well known that classifier-free guidance can produce distributional shifts. Have you tried alternative guidance (e.g., from Karras et al., 2024) to see if it pairs well with MSD?
2. In practice, the authors rely on partial integration heuristics and limiting guidance to certain intervals. These are effective but might appear less principled. I guess additional exploration of robust integrators or dynamic bandwidth strategies could improve this.
3. As I mentioned above, this paper focuses heavily on SDS vs. MSD, plus SDI. While that is a fair baseline set, the discussion might be enriched by direct comparisons with specialized variance-reduction or novel guidance techniques.
4. In text-to-3D tasks, you mention 7k steps. Does the improved stability substantially reduce the required number of iterations or does it primarily improve final fidelity?

**Questions For Authors:**

Please refer to the "Weaknesses" section.

**Relation To Broader Scientific Literature:**

Existing “score distillation sampling” approaches like SDS are widely used in text-to-3D tasks, but often criticized for high variance, poor convergence, and potential for spurious modes. The proposed mean-shift view is a novel perspective, linking classical KDE-based ideas to diffusion-based generative modeling. In this paper, the authors situate their work among mainstream diffusion references (Song et al., Karras et al., etc.), and they cite methods for variance reduction or improved multi-step approaches in SDS.

**Theoretical Claims:**

The authors leverage standard results from kernel density estimation and mean-shift. The proofs or derivations that show how the kernel’s gradient aligns with the data distribution’s gradient are standard but well-structured here. The demonstration that a product density approach (combining kernel $G_{\lambda}$ with $p$) can be sampled through a modified diffusion score is logical and consistent with known properties of diffusion sampling.

---

> ### Author Rebuttal · Authors · 2025-03-31
>
> We thank the reviewer for the thoughtful and constructive review. We are glad they recognize our contributions. Below, we answer questions raised in the review:
> > Comparisons with other guidance techniques and variance-reduction methods.
>
> **Guidance**. In low-dimensional settings (eg, our toy experiments), our method can recover the modes and reconstruct the data distribution well without any guidance (See Fig. 1). This is aided by the fact that the conditional score estimates parameterized as $\epsilon_\theta(z_t, c)$ (predicted noise from the pre-trained network) is good by itself, without guidance i.e. $\tilde{\epsilon}_\theta (z_t, c)$. Empirically, we observe that without guidance, ancestral sampling techniques like DDIM produce samples that lie on the data manifold, albeit with few outliers (See Fig. 2).
>
> This is not the case in the high-dimensional setting with experiments on Stable Diffusion. Here $\epsilon_\theta(z_t, c)$ samples are noticeably bad and are predominantly outliers. Currently, the best fix is to augment these noise estimates with guidance to produce $\tilde{\epsilon}_\theta (z_t, c)$, the strategy prevalent in sampling algorithms. We inherit these practices when performing distillation.
>
> We recognize the existence of alternative guidance techniques to CFG, like Autoguidance (Karras et al., 2024). As these methods pair well with DDIM (and other ancestral sampling techniques), we believe the benefits will extend to distillation-based methods like ours. Ultimately from the perspective of distillation, different guidance simply changes the shape of the output distribution but does not fundamentally change the mechanics of diffusion. We used CFG in all our experiments as it is more widely used, has hyperparameters (guidance scale) that have been more rigorously tested by the community, and was used in all our baselines (SDS and SDI).
>
> We extend **Table 2** with comparisons between two guidance techniques, CFG vs Autoguidance (Karras et al., 2024). We add a new baseline VSD [1] (suggested by Reviewer cVvL) and our original baseline SDI. We show just the fractal dataset.
>
> Please see the **rebuttal to Reviewer cVvL** for additional comparisons with VSD.
>
> Left: *learned denoiser with CFG* / Right: *learned denoiser with Autoguidance*. Best performance among distillation-based methods is highlighted in **bold**.
>
> |Dataset|Method|NLL↓|Precision↑|Recall↑|MMD↓|
> |-|-|-|-|-|-|
> ||DDIM†|-1.59/-1.67|0.97/0.96|0.44/0.79|257.43/0.25|
> ||SDS|15.96/11.33|0.17/0.04|0.03/0.03|3875.11/71.05|
> |Fractal|VSD|18.97/11.52|0.21/0.03|0.05/0.03|3845.41/**70.25**|
> ||SDI|27.375/0.652|0.30/0.51|**0.48/0.51**|**69.23**/15089.58|
> ||Ours|**-1.15**/**-1.99**|**0.94**/**0.97**|0.40/0.43|133.41/122.94|
>
> **Variance reduction**. One of the variance reduction methods we cite, SteinDreamer [2], propose using control variates to minimize the excessive variance present in SDS. This excessive variance comes from the randomly sampled noise added to the target (or rendered images). Unlike SteinDreamer, SDI finds a better approximation of this desired noise term, eliminating one of the root causes of the excessive variance instead of compensating for it later. We believe this makes SDI a better candidate among previous variance reduction methods. Moreover, the official implementation for SteinDreamer is not yet publicly available. We will be happy to make comparisons as soon as an official implementation gets published. We would be happy to add comparisons to any other specific work the reviewer has in mind.
>
> > The impact of integration heuristics and their ad-hoc nature, additional exploration of robust integrators or dynamic bandwidth strategies.
>
> We tried robust integrators in toy settings, and while they do indeed improve numerical stability, the additional score model evaluations they require make them impractical in a higher-dimensional setting. Higher-order numerical solvers like PNDM [3] did not improve numerical stability in our setting. We observed limited guidance to be the simplest to implement and most impactful. Adaptive bandwidth strategies have been proposed for Kernel Density Estimation in the past [4]. We hope to extend these techniques for our use case in the future.
>
> > Does the improved stability substantially reduce the required number of iterations, or does it primarily improve final fidelity?
>
> Yes, the improved stability reduces the number of optimization iterations while maintaining the fidelity of the output.
>
> References:
>
> [1] Wang Z, Lu C, Wang Y, et al. “Prolificdreamer: High-fidelity and diverse text-to-3d generation with variational score distillation”, 2023.
>
> [2] P. Wang et al. “Steindreamer: Variance reduction for text-to-3d score distillation via stein identity”, 2023.
>
> [3] L. Liu et al. “Pseudo Numerical Methods for Diffusion Models on Manifolds”, 2022.
>
> [4] D. Comaniciu et al. “The Variable Bandwidth Mean Shift and Data-Driven Scale Selection”, 2001.

---

### Official Review · Reviewer_cVvL · 2025-03-18

**Overall Recommendation:** 2

**Summary:**

This paper introduces mean-shift distillation, which improves the convergence behavior of the SDS objective. Experiments on synthetic and real datasets demonstrate the effectiveness of the proposed method.

**Claims And Evidence:**

Yes.

**Essential References Not Discussed:**

No.

**Experimental Designs Or Analyses:**

Yes. I've checked the soundness and validity of all the experiments.

**Methods And Evaluation Criteria:**

Yes.

**Other Comments Or Suggestions:**

1. Missing $\sigma_t$ in Eq. 12.

2. Table 2 does not contain a comparison between an ideal and a learned denoiser.

3. There are many typos in Algorithm 2. For example, the indices in line 560 and 568, $z_T$ in line 569. In line 577 to 579, it seems that $y_1 = y_2 = \cdots = y_N$.

**Other Strengths And Weaknesses:**

Strengths:

The proposed gradient proxy improves the SDS objective by reducing variance and enhancing mode alignment.

Weaknesses:

1. Writing Clarity: This paper requires significant revision for clarity. The notations are particularly confusing. For example, $y$ represents the text embedding in the right column of line 70, but is also used for data samples in Eq. 2, and has a different meaning in Eq. 14. Additionally, Eq. 9 is unclear as $z_t$ is not defined. There are lots of undefined notations between Lines 39 and 45.

2. Baseline Comparison: The baseline appears relatively weak, as SDS is an early work in this field.

**Questions For Authors:**

1. How are the losses in Algorithms 1 and 2 derived?

2. Since the variational score distillation (VSD) loss developed by Prolificdreamer [1] is widely used for distilling diffusion models in image generation [2], how does the proposed method compare to the VSD?

Reference

[1] Wang Z, Lu C, Wang Y, et al. Prolificdreamer: High-fidelity and diverse text-to-3d generation with variational score distillation[J]. Advances in Neural Information Processing Systems, 2023, 36: 8406-8441.

[2] Yin T, Gharbi M, Zhang R, et al. One-step diffusion with distribution matching distillation[C]//Proceedings of the IEEE/CVF conference on computer vision and pattern recognition. 2024: 6613-6623.

**Relation To Broader Scientific Literature:**

The mean-shift distillation developed in this paper may provide a new tool for 3D generation as well as one-step text-to-image generation.

**Theoretical Claims:**

N/A.

---

> ### Author Rebuttal · Authors · 2025-03-31
>
> We thank the reviewer for recognising our contributions. We acknowledge the feedback regarding our notations.
> ## 1 Writing clarification.
> As we cannot update the submission, we provide them below. We believe these are minor corrections and will incorporate them in our revision.
> |Notation|Clarification|
> |-|-|
> |$y$ in line 70| In image diffusion models, $y$ is the common convention for text prompt. Alternatively, $c$ is used for any conditioning signal. We will use the latter.|
> | $y$ in Eq. 2, Eq. 14| In mean-shift and KDE, $y_i$ can denote samples from product density or successive locations of kernel $G_\lambda$ [Comaniciu & Meer, 2002]. In Eq. 2 we use the same name for the integration variable of the convolution so that we would match this convention once we discretize the integral. Karras et al. (2022) use $y_i$ for data samples. We recognize the confusion.|
> |$z_t$ in Eq. 9| Line 158, right column, we use the product sampling trick from Song et al., 2021b; Dhariwal & Nichol, 2021. Here, $x_t$ or $z_t$ refers to the noisy sample or noisy latent, respectively. We will explicitly define this.|
> |Algorithm 2, indices (line 560, 568) and $z_T$ (line 569)| We acknowledge this is not standard notation for discretizing $t$ and the typo in line 569. We will change indices to $t = T, …, 1$ and change $z_t$ in line 561, 569 to $z_{t-1}$|
> |undefined notations between Lines 39 and 45| This paragraph is intended to define those notations, but contains one typo; $g$ and $M$ are supposed to be the same.|
> |Line 577 to 579, is $y_1=…=y_N$?| While $x^k$ and $\lambda$ are the same, there is stochasticity in the solver. Initial $z_T$ is either randomly initialized (naive) or initialized via inversion (stable). The latter is not deterministic and can have numerical errors. Denoising process is also not deterministic. Thus, each $y_i$ is different, and averaging them gives us unbiased estimate of the gradient. We will make stochasticity explicit.|
> > How are losses in Algorithms 1 and 2 derived?
>
> Score distillation methods (SDS, SDI, Ours) provide gradients of the loss w.r.t target parameters but not the loss itself (see Appendix A.4 in DreamFusion (Poole et al., 2022)). Losses are given implicitly. In our toy experiments, we reconstruct the loss function by integrating finite differences and visualizing them in Fig. 1, 2, and 3. Section 4.2 provides more details.
>
>
> ## 2 Additional baselines.
> We compare with VSD (PriolificDreamer, 2023) below. Note the following about VSD:
> - VSD claims (Appendix C.3) SDS's mode-seeking causes over-saturated, low-diversity results. We find SDS is not mode-seeking; its high variance and bias from modes are the reasons for poor results. See our rebuttal to Reviewer 6KVp on why mode-seeking is desirable for distillation.
> - VSD needs task-specific fine-tuning (via LoRA or training U-Net) to estimate the variational score $\nabla_{z_t} \log q^{x}_t$. Our method needs no fine-tuning, and it’s unclear how VSD’s approach generalizes across domains.
> - SDI outperforms prior works like VSD in text-to-3D generation. SDI is also easily extensible to any domain, making it a strong baseline.
>
> We extend **Table 1** comparing ideal and learned denoiser. We add new baseline VSD and original baseline SDI.
>
> Left: *ideal denoiser* / Right: *learned denoiser*.
>
> Efficiency ($\uparrow$).
> |Dataset|SDS|VSD|SDI|Ours|
> |-|-|-|-|-|
> |Fractal|-7.4/-6.9|-5.9/-3.8|14.2/14.2|13.4/7.7|
> |Spiral|-8.5/-7.6|-6.8/-4.4|13.9/14.2|13.4/6.3|
> |Pinwheel|-7.8/-7.0|-6.4/-3.9|14.2/14.2|13.8/7.1|
>
> Similarly, we extend **Table 2**. Best performance among distillation-based methods in **bold**.
> |Dataset|Method|NLL↓|Precision↑|Recall↑|MMD↓|
> |-|-|-|-|-|-|
> ||DDIM|-1.85/-1.51|0.97/0.95|0.93/0.96|0.86/0.007|
> ||SDS|36.15/9.12|0.08/0.01|0.03/0.0|328.03/87.04|
> |Fractal|VSD|9.97/9.88|0.05/0.10|0.02/0.05|230.97/94.68|
> ||SDI|24.28/**-2.87**|0.27/**0.97**|0.01/0.12|**29.927**/459.89|
> ||Ours|**-1.32**/-2.02|**0.92/0.97**|**0.33/0.42**|30.46/**12.79**|
> |||||||
> ||DDIM|-1.39/-1.32|0.97/0.96|0.93/0.96|0.41/1.16|
> ||SDS|30.37/8.13|0.02/0.04|0.03/0.11|13.85/274.35|
> |Spiral|VSD|10.15/8.90|0.04/0.07|0.09/0.14|23.46/271.84|
> ||SDI|35.64/19.16|0.1/0.12|**0.9/0.42**|39.51/2008.3|
> ||Ours|**-1.28/-1.51**|**0.99/0.98**|0.18/0.18|**4.49/18.41**|
> |||||||
> ||DDIM|-1.19/-1.1|0.97/0.97|0.94/0.97|1.05/0.27|
> ||SDS|2.29/2.00|0.85/0.90|0.03/0.005|**5.18**/36.37|
> |Pinwheel|VSD|3.34/2.28|0.65/0.97|**0.04**/0.019|6.78/33.36|
> ||SDI|28.31/17.33|0.17/0.51|0.001/**0.15**|6.13/98.09|
> ||Ours|**-1.94/-2.19**|**0.99/0.99**|0.01/0.13|5.83/**7.25**|
>
> Note: With ideal denoiser, the efficiency of our method is comparable to SDI. With learned denoiser, SDI has better efficiency. Despite this, we produce better samples.
>
> Similarly, we extend **Table 3**.
> |**Method**|**FID↓**|**CLIP-SIM(L/14)↑**|
> |-|-|-|
> |DDIM|-|44.1±2.8|
> |SDS|199|27.7±1.9|
> |VSD|158|30.8±1.4|
> |SDI|166|31.0±0.7|
> |Ours|114|32.6±0.8|
>
> We accompany Table 3 with additional qualitative comparison for text-to-2D: https://imgur.com/a/BWpHde5.

---

### Decision · Program_Chairs · 2025-05-01

**Decision:**

Reject

**Comment:**

The paper introduces a mean-shift mode seeking-based diffusion distillation technique, which is applied to both text-to-image and text-to-3D applications. In comparison to Score Distillation Sampling (SDS), the proposed mean-shift distillation achieves better mode alignment and lower gradient variance, leading to improved performance. However, the practical value of this method needs further demonstration:

- To establish mean-shift distillation as a foundational technique for diffusion distillation, it would be helpful to validate it on unconditional diffusion models. Notably, Figure 1 of the paper, drawing inspiration from Karras et al. (2024) who used a similar setup to motivate their work, "Guiding a diffusion model with a bad version of itself," raises the question of why distillation of unconditional diffusion models, the key task explored by Karras et al. (2024), was not considered in this study.

 - While mean-shift distillation appears effective in addressing the limitations of SDS, it is unclear why it has not been compared with or combined with VSD (also known as Diff-Instruct or DMD in text-to-image generation). These methods improve upon SDS and are often considered stronger baselines for diffusion distillation, particularly for unconditional, label-conditional, and text-to-image diffusion models.

 - Although the authors included VSD in their baseline during the rebuttal, the results were primarily on toy examples rather than widely used benchmarks for evaluating diffusion models and their distilled versions.

  - The AC would be more supportive of this paper if the authors demonstrated that their technique remains effective when applied to VSD, Diff-Instruct, or DMD, and/or conducted a comprehensive evaluation of their method on more well-recognized benchmarks for diffusion models and diffusion distillation.